# Identifying Māori perspectives on gene editing in Aotearoa New Zealand

Amanda Clark, Phillip Wilcox, Sandy Morrison, Debashish Munshi, Priya Kurian, Jason Mika, David Chagne, Andrew Allan & Maui Hudson

Māori perspectives on gene technologies are evolving, and traditional cultural constructs continue to inform a wide diversity of views. Here we summarise a series of research activities aimed at identifying evolving Māori perspectives on gene editing and how these inform engagement at the co-innovation interface.

Research on gene-editing technologies is advancing rapidly in biomedical and agricultural laboratories around the world, especially as the advent of the CRISPr-Cas9 has been described as "a Midas touch for researchers editing genomes"[1]. So ubiquitous is this research that *Communications Biology* alone has published over 350 articles (and growing) on the applications of this technology since 2018 (see, e.g., Li et al.;[2] Gopalakrishna et al.;[3] Schubert[4]). Yet, public perceptions of gene editing are still largely polarised because of adversarial messaging by proponents, who talk up the revolutionary potential of better treatment of human diseases and the production of better crops, and opponents, who focus on the risks of these new technologies[5]. Although public engagement on gene editing is on the rise[6,7], significant demographic groups that have not been adequately factored into such exercises are Indigenous people in different parts of the world. As "key stakeholders," Indigenous people need to be consulted actively to confront and subvert "power imbalances that marginalize Indigenous ways of knowing" (p. S74)[8]. We address this gap by empirically identifying the perspectives of Indigenous Māori communities of Aotearoa New Zealand (AoNZ) on gene editing.

Māori perspectives on gene technologies in the contemporary environment are evolving, and traditional cultural constructs continue to inform a wide diversity of views[9–11]. The historic mistrust associated with genetic modification (GM)[12] has softened in recent decades to accommodate more case-dependent engagement. Incorporation of Māori values in national regulation is perceived as key to a more balanced integration of biotechnologies in both cultural and socio-economic contexts[13], and a growing body of literature addresses the need for more agile, participatory regulation in the biotech era[14–16]. There is also an increasing recognition of the transferability of Māori-focused approaches to biotechnologies to other Indigenous communities and society in general[17].

Acknowledging that "communication on new technologies requires attention not merely to science and economics, but also to culture" (p.7)[18], this paper responds to the 'what next' for the culturally appropriate integration of genomics and gene editing in AoNZ. More specifically, it summarises the findings of a multi-level research project carried out at a Co-Innovation interface, describing engagement with Māori worldviews, at the nexus of cultural and commercial interests. The paper refers to genomics as well as gene editing as both technologies are used in the country (for example, in the horticultural sector to inform cultivar breeding decisions) and there are similarities in the way Māori relate to them.

A diverse team of inter-disciplinary researchers spanning biological and social sciences were involved in multiple projects to gauge Māori perspectives on the implications of using genomics and gene editing to speed up plant breeding for selection of improved plant cultivars. A series of workshops, interviews and surveys engaged with over 1000 individuals - from grassroots communities to (Māori and non-Māori) academic and business spheres, and Māori scientists working at the interface of genomics and gene editing. The analysis builds on a rich corpus of data, including an extensive review of publications exploring Māori attitudes to genomics and gene editing[19], scientific research in cultural contexts[20], as well as a national survey of a stratified random sample of an equal number of Māori and non-Māori respondents[18]. Further insights came from an international Indigenous Genomics Conference which attracted over 130 indigenous academics and scientists whose presentations addressed issues as diverse as engagement with regulatory bodies, biobanking indigenous DNA, and using gene editing to eradicate exotic pest species.

## Methods

The research adopted a mixed-method approach incorporating qualitative interviews and workshops as well as a quantitative survey to capture the diversity of Māori perspectives on gene editing from across the age and experience spectrum. The study was approved by the Te Manu Taiko Ethics Committee at the University of Waikato and all ethical regulations relevant to human research participants were followed including gaining informed consent. The project began with a scoping exercise, involving a literature review and pilot interview-based survey[19], followed by a Summer Internship for Indigenous Peoples in Genomics (SING), and interviews with professionals at the interface of Te Ao Māori (the Māori world) and science, to identify the implications of genomic research for Māori communities (2019–2020). Alongside the interviews, the research team carried out a national survey of 830 Māori and non-Māori respondents to analyse 'how the diversity of values of a wide range of stakeholders on gene editing technologies [can] be negotiated to shape a robust policy design on the use and regulation of these technologies in a variety of sectors and contexts'[18].

The cultural aspect of the research was reinforced by: active engagement with Indigenous perspectives on the intersections of science, business, and Indigenous knowledge systems at a Māori thought-leaders workshop on genomics, taonga species and emerging commercialisation prospects (September 2019); an interactive session on gene editing at an Indigenous Genomics Conference (January 2020); a cultural *wānanga* ('sense-making workshop') in which cultural experts and community members considered how learnings from the research might be utilised in policy and practice (September 2020); and a primary sector *hui* (meeting, gathering) of

representatives of Māori entities operating in the primary sector (forestry, horticulture, agriculture), to explore views on how gene editing and research could be commercially applied (December 2020). Each of these activities followed Māori protocols and created an appropriate environment for Māori participants to share their thoughts openly and freely.

The outcomes of all the research engagement activities were collated, analysed thematically, compared, and synthesised to illustrate the rich nuances of Māori perspectives on genomics and gene editing. The cultural robustness of our method was in the iterative discussion of results and collaborative sense-making across the series of activities. While this remains qualitative, many other aspects are quantitative, as replicates and numbers show.

**Reporting summary.** Further information on research design is available in the Nature Portfolio Reporting Summary linked to this article.

## Results

**Literature review and pilot interviews/survey.** Three activities were the foundation of this (previously published) research element:[19]

(i) A review of 38 peer-reviewed papers about Māori perspectives on biotechnologies and the wider field of genetics published between 2005 and 2017,

(ii) Informant interviews ($n = 8$), and

(iii) A pilot survey ($n = 9$) with Māori stakeholders and individuals.

The literature review identified that Māori are more likely to be positioned on the 'anti-GM' end of the spectrum[21–23]. The review also identified key Māori cultural concepts and values relevant to Māori views on biotechnologies and genetic research: *whakapapa* (genealogy), *mauri* (life essence), *mana* (power/authority), *kaitiakitanga* (guardianship), *mātauranga* (indigenous knowledge), *tikanga* (protocols), *Papatūānuku* (earth mother) and *tangata whenua* (indigenous people/'people of the land'). Some other cultural concepts referred to included *kawa* (customary principles), *tika* (right/correct), *manaakitanga* (to care for, look after), *tapu* (sacred/restricted), *taonga* (precious), *wairua* (spirit) and *tākoha* (gift)[19].

Participants in the pilot interview and survey did not oppose new and emerging gene editing technologies, per se, but raised concerns about how they would be used. The experience of the participants played an interesting part in the identification and proposed management of potential risk. Those with a background in a particular sector (e.g., the environment) were more comfortable employing gene editing in that domain and highlighted risks associated with other areas (e.g., health), and vice-versa. Participants shared a sense that there will always be both justifiable and unpalatable use cases. A dynamic approach to regulation was generally preferred, with specific applications being approved on a case-by-case basis. Participants emphasised that such approvals should align with Māori values and prioritise community benefit over commercial interests.

Reflecting on the role of Māori values and cultural concepts in guiding Māori perspectives on future regulation of biotechnology, participants felt that the concepts of whakapapa, mauri, mana and kaitiakitanga could provide cultural 'grounding' for consideration of gene editing's ethical dimensions. Incorporating Māori values into decision-making processes could provide a balancing factor, to ensure broader community interests were a key consideration in future uses of gene editing technologies[19]. Māori values could be enhanced or diminished by gene editing projects depending on the context and level of control in the project.

**SING Aotearoa 2019 workshop.** At a Summer Internship for Indigenous Peoples in Genomics (SING) programme over five days in 2019, 20 interns explored topics ranging from genetics, gene editing, bioinformatics and data sharing, to incorporating tikanga practices in the laboratory, access and re-use policies, commercialisation, partnering with science, genetic research on taonga species and 'future thinking' about breeding technologies in AoNZ.

The Future Thinking workshop provided an opportunity to contemplate practical prospects of gene technologies in a range of inter-cultural contexts. Interns formed groups to consider four different case-study scenarios. Each scenario comprised four critical elements: (i) a 'personnel types' relationship, (ii) a breeding technology, (iii) a genetic engineering intervention and (iv) a resulting outcome. For example, one scenario involved (i) 'a Māori entity working with a research institute', (ii) using 'traditional breeding' technologies and (iii) 'gene editing', with (iv) the outcome of 'a new medicine based on an Indigenous species.'

Interns identified a range of issues with the scenarios, including cultural factors, economic factors, health impacts, access-and-benefit-sharing (ABS), control over data and Intellectual Property (IP), as well as the responsibilities of *kaitiaki* (traditional stewards). These responses were then available for 'endorsement' by other interns, with the endorsed responses collated and thematically analysed within and across the range of scenarios. Overall, groups concurred that gene editing had potential to be either positive or negative, largely depending upon relationship management, the values espoused, and the level of participation of and control for Māori entities.

**Practitioner interviews.** Interviews with seventeen key informants addressed Māori participation in the commercialisation of genomics research. Drawn from a spectrum of exposure to the Co-Innovation interface, participants ranged from non-Māori genomics scientists to Māori and non-Māori working in science-based or community-liaison roles and Māori working in genomics, law or business. All had been involved in scientific research and had some knowledge of genomics. Over half were Māori, strong in tikanga as well as professional expertise[19].

Identified issues included systemic exclusion of Māori from processes and policies around the ethics of data acquisition, storage, interpretation and use/re-use, with active concern for bio-piracy, ABS and IP protection. Potential benefits included the creation of new mātauranga, health improvements, economic opportunities (including IP development) and enhancement of Māori cultural values.

A central theme that emerged was the need for trust, based on strong, enduring relationships conducted with transparency and integrity. Both Māori and non-Māori interviewees advocated for inclusion of Māori values of *whakawhānaungatanga* (relationship building), *manaakitanga* (care) and *kaitiakitanga*, in research and commercialisation.

None of the informants were strongly pro-commercialisation. Most agreed there were huge potential benefits to be gained from genomic research but emphasised 'who stands to benefit should always be front of mind'. Concerns included cultural and environmental impacts as well as risks from unscrupulous human interests. There was a clear message of caution and the need to ensure transparency and accountability for all involved parties. A repeated suggestion was that 'genomics' might simply lack priority for many Māori 'who have more urgent matters to deal with'. There was recognition of the realistic and necessary time investment and potentially long delays until benefits result from research.

While somewhat cynical about historic commercialisation, informants expressed interest in engaging with genomic research and commercialisation independently of universities or Crown research institutions. Capacity building (especially among youth) was perceived as key to this, potentially through standalone Māori research institutions. Informants made positive

reference to genomic projects such as the Miro blueberry enterprise (https://www.miroberries.com/) and the work of Hikurangi Enterprises (https://hikurangi.enterprises/).

Overall, this research-aware group was circumspect, but not negative, about prospects for Māori engagement with the commercialisation of genomic research outcomes. Their concerns were based on working knowledge of the issues around co-innovation and provided keen insights to the everyday Māori world and its potential appetite for engaging with new scientific research. Their qualified willingness to engage in future genomics-based research, such as gene editing, stemmed from recognition of a need for Māori ownership of new mātauranga and the fact that modern gene technologies are generally perceived as being more acceptable than traditional genetic modification.

**The national survey**. A national survey on *"Mapping Values, Beliefs, and Attitudes on Genetic Technologies"*[18] surveyed a stratified random sample of an equal number of Māori and non-Māori participants ($n = 830$) to provide a snapshot of the beliefs, values, and attitudes of Indigenous and non-Indigenous people in AoNZ towards genetic technologies. The ratio of Māori respondents was oversampled to facilitate direct comparison of Māori and non-Māori perspectives on new genetic technologies. The survey explored similarities and differences in perspectives between Māori and non-Māori participants about gene technologies, and patterns in the support of or opposition to gene technologies. Respondents also provided insights on their awareness of specific gene editing technologies; attitudes to a range of different uses of genetic modification and gene editing; support for current legal frameworks; and perspectives on the role of Māori values in providing guidance on the use, control, regulation, and commercialisation of gene editing.

The development of typologies using K-Means cluster analysis identified six distinct clusters – three Māori clusters (*Strongly Supportive, Leaning Supportive*, and *Strongly Opposed*) and three non-Māori clusters (*Strongly Supportive, Leaning Supportive*, and *Opposed*). Although the quantitative data showed that 79% of the sample population supported or were open to supporting gene technologies, the open-ended responses to questions showed much greater nuance and complexity. Over half of the total respondents (56 per cent) who were in the middle clusters (*Leaning Supportive*) provided a quantitative score that leaned closer toward support for the technologies, but their open-ended responses found strong ambivalence due to the uncertainty they felt about benefits and risks.

Interestingly, the percentage of respondents in support of gene technologies was similar in both Māori and non-Māori groups but more Māori were opposed to such technologies than non-Māori; "indeed, those Māori who are opposed are not just opposed but are "strongly" opposed" (p. 5)[18]. A majority of respondents, both Māori and non-Māori, identified 'taonga species' as an important Māori value, followed by 'kaitiakitanga', consistent with findings from Research Elements I and III above. There was also alignment between Māori strongly supportive, non-Māori strongly supportive and non-Māori leaning supportive in relation to 'pro-commercialisation' terms, which associated 'greater good', 'equitable ABS', and 'consultation and protection of rights' as the most important considerations. Support for commercialisation revealed the greatest variability between the groups, ranging from 81–85% in the strongly-supportive groups, to 50–55% in the leaning supportive and 24–25% in the opposed groups.

The survey findings indicate that, in general, despite continuing scepticism about scientific legitimacy, certainty of outcomes, concern for environmental impacts, genuine cultural recognition, engagement and rights protection, there was considerably less opposition to gene editing than to traditional GM. It is generally accepted that gene editing offers an increased level of control that, while not absolute, provides more defined benefits and clarity about risks. For the majority of Māori surveyed, these risks and benefits were not incompatible with Māori values. Also noteworthy was broad non-Māori endorsement of Māori values in this context, perhaps reflecting the increasing societal adoption of Māori language and concepts in AoNZ.

**Māori thought-leader workshop**. A workshop in 2020 for 30-plus Māori thought-leaders and interested stakeholders briefed participants on developments in the fields of genomics and gene editing involving taonga species. Feedback was sought on the values that should underpin the meeting of taonga species and modern gene technologies. Discussion centred on the roles and responsibilities of kaitiaki in relation to taonga species, and the differences between kaitiaki relationships for taonga species, *mātauranga taketake* (associated traditional knowledge), and the physical samples and data generated from the taonga.

Using different case-study examples, participants debated the nature of kaitiaki relationships, the range of 'rights and interests', and potential mechanisms to support greater Māori control of data and IP. Feedback included: that benefits must include non-economic values; agreements should include shared goals and aspirations; that even when Māori-led, projects required 'big conversations' to maintain spiritual integrity; and that targeted communication is critical to appropriate engagement. Reservations were expressed regarding justifications for certain gene editing applications (ie, editing the colour of animal skins to endure global warming effects) and the need for transparency around failures as well as successes was identified. Similar to the workshop and survey, concerns included: 'What are we not being told?' 'What are the adverse effects?' 'Who owns and controls the technology?' 'Who is doing the research?' and 'Is gene editing reversible?'

Overall, the feedback from this workshop was high-level but 'control' oriented, providing more precise statements than other groups about legislation, industry codes, international alignments and genomics ethics. The group was overtly sceptical about gene editing but nonetheless anxious to achieve practical, community-level engagement with, and ownership of, genomic knowledge and data. Transparency and accountability were identified as key objectives for all parties, at all levels and stages.

**SING Conference 2020**. The 2020 SING Indigenous Genomics 'Gene editing' session showcased diverse Māori and Indigenous perspectives. Themes included: use of Māori knowledge in Royal Society of New Zealand (RSNZ)'s scenario-based gene editing examples of healthcare, pest control and primary industries;[24,25] gene-drives to protect taonga species as 'culturally significant flora and fauna species'; and a mānuka genome sequencing project[26,27]. A more reciprocal model of engagement was proposed, where unidirectional 'science communication' is not conflated with participatory community engagement, and 'consent' is not a guaranteed outcome of partnership.

**Cultural wānanga**. The purpose of this 2021 wānanga was to engage cultural experts and community members in an exercise of collaborative 'sense-making'. Results from the above Research Elements were shared with participants, who discussed how Māori values, concepts, and modes of thinking can support decision-making as we navigate uncharted territory. Often Māori words like kaitiaki are used as a 'placeholder' to provide 'space for thinking'. Participants supported the views emerging from previous research activities and identified a need for more regular spaces to continue dialogue, and to consider application or implementation around specific domains and/or specific projects. An initial set of principles to guide gene editing decision-making was proposed:

(i) Me āta haere – proceed cautiously.

(ii) Kaua e haere i te rorirori – be motivated by good intent.

(iii) Whaia nga tohutohu o ngā rangatira – follow the guidance of cultural experts.

(iv) Me hīkoi tahi te mātauranga me te pūtaiao – utilise mātauranga and science.

(v) Mātiro whakamua – consider future consequences.

(vi) He kōrero hokia – keep people informed.

Cultural experts expressed an interest in continuing discussions around the appropriate use and application of gene editing technologies.

**Primary sector workshop.** The primary industries account for about one-third of the burgeoning Māori economy. This 2021 workshop brought together a group of (mostly forestry) industry stakeholders and practitioners, to consider the potential of gene editing. Issues of mātauranga Māori and ABS are routinely afflicted by tokenistic appropriation of Te Ao Māori concepts and institutional acquisition of Māori IP. These historic norms were reviewed in light of the implications of technologies such as gene editing. The coronavirus pandemic provided timely additional context, raising concerns for food quality and security and the need to return to more holistic, circular economies with '*whenua* (earth) first' principles.

Concluding that pan-Māori commonalities are greater than regional or inter-tribal differences, the discussion then turned to some of the practical implications arising from genomic technologies including:

(i) **Relationship reinvention:** the inadequacy of the Treaty of Waitangi 'consultation' (*cf* partnership) model and the need for ongoing negotiation and collaboration, in both Māori-Government and Māori-Māori spheres.

(ii) **Tikanga values prioritisation:** the need to ensure an 'integral ecosystem' perspective is maintained.

(iii) **Genomic literacy:** the immediate need for pervasive, accessible, relevant community education.

(iv) **Data management:** the need for a framework that collates/translates kaitiakitanga and science metrics, providing for risk analysis and decision-making.

(v) **'Genomic sovereignty':** the prospect of a Māori-initiated and run genomic research facility.

(vi) **Succession planning:** strategies and structures are needed to ensure *rangatahi* (young people) are properly prepared for the challenges they will face in this sphere.

(vii) **Funding:** investment levels, sources, and priorities must be identified.

(viii) **Ownership**: demonstrated through whakapapa – and the 'meaningful partnership *tohu*' (symbol) of (an expanded) WAI262.

(ix) **The time imperative**: there is immediate need for practical, committed leadership and mechanisms for acceleration of decisions on the use of genomic technologies.

Gene technologies thus present a critical opportunity for an enhanced Treaty relationship, with pan-Māori commitment and whenua-first principles contributing to decision-making that protects the environment and benefits Māori and wider AoNZ. Whenua-first relates to prioritising the sustainability of land and/or natural resources as a foundational element supporting commercial outcomes. Contemporary interpretations of the Treaty expect greater participation and engagement with Māori, including co-governance in some instances, in the development of policy and use of natural resources[28].

## Discussion

The Research Elements described in this paper represent the most comprehensive collection of qualitative and quantitative data about Māori perspectives on gene editing to date. Reflective of the nature of debate and discussion in AoNZ (and generally), including the recent Royal Society of New Zealand consultation process[29], key outcomes represent evolving Māori understandings and positionality. It is important to reiterate that Māori perspectives represent a broad range of views, with Māori both supporting and opposing gene editing.

Figure 1 summarises this breadth of views - acknowledging the potential of genetic technologies such as gene editing, providing they are employed within a framework of Māori principles and values, and a culture of bi-directional knowledge-sharing and capacity building. Recognition of potential is balanced by scepticism regarding 'control' and cost-benefit issues. Gene editing 'could go either way' from both scientific and cultural perspectives. Regulation needs to recognise that gene editing is 'uncharted territory' with potential to add to (rather than replace) existing approaches, and Māori values provide holistic modes of thinking for its navigation.

While there is general support for a cautious approach, what this looks like, from a sector, area, or iwi-based perspective remains unclear. However, taken as a whole, the key messages reinforce a maturing and nuanced conversation that is open to exploring how gene editing might contribute towards delivering positive outcomes for Māori communities and businesses, and reiterate 'the importance of addressing the continuing influence of [historic perspectives]'[19].

Regulation of GM technologies has long been informed by the Precautionary Principle or a precautionary approach. Developed at the 1992 Rio Earth Summit, the principle obliged member States, when contemplating potential adverse effects to the environment, to take precautionary measures, even where no scientific proof of such potential damage was available. AoNZ's Hazardous Substances and New Organisms Act 1996 ("HSNO") codified the obligation to "…take into account the need for caution in managing adverse effects where there is scientific and technical uncertainty about those effects" (s7)[30].

Globally, the principle is being re-defined. Perceived as an obstacle to innovation, guidance increasingly promotes 'solutionist' proactivity. In AoNZ, the draft Natural and Built Environments Act (NBEA) describes precaution as:

'… an approach that, in order to protect the natural environment [from] threats of serious or irreversible harm … favours taking action to prevent those adverse effects rather than postponing action on the ground that there is a lack of full scientific certainty.'

While semantically 'positive', there is scope for the revised approach to operate in the tradition of ABS, whereby developers offset environmental or long-term costs by providing short-term material or otherwise unrelated 'gains' to affected communities. Whole-of-ecosystem frameworks (as per the Sector workshop) could assist protective, values-based decision-making.

The RSNZ Report[31] observes that 'process based regulatory systems … will become increasingly obsolete and unsustainable' proposing instead a 'risk-tiered approach', as per that being developed in Australia (Legislative and Governance Forum in Gene Technology)[32]. While the proposed agility and proportionality of oversight in the Australian model may appeal to regulators, it also pursues 'broader environmental release of genetically modified organisms and … gene-drive organisms' as a fixed goal, with no evident cultural-analysis ingredient[32].

The Royal Society of New Zealand conclusions below are consistent with the themes that emerged from our research activities:

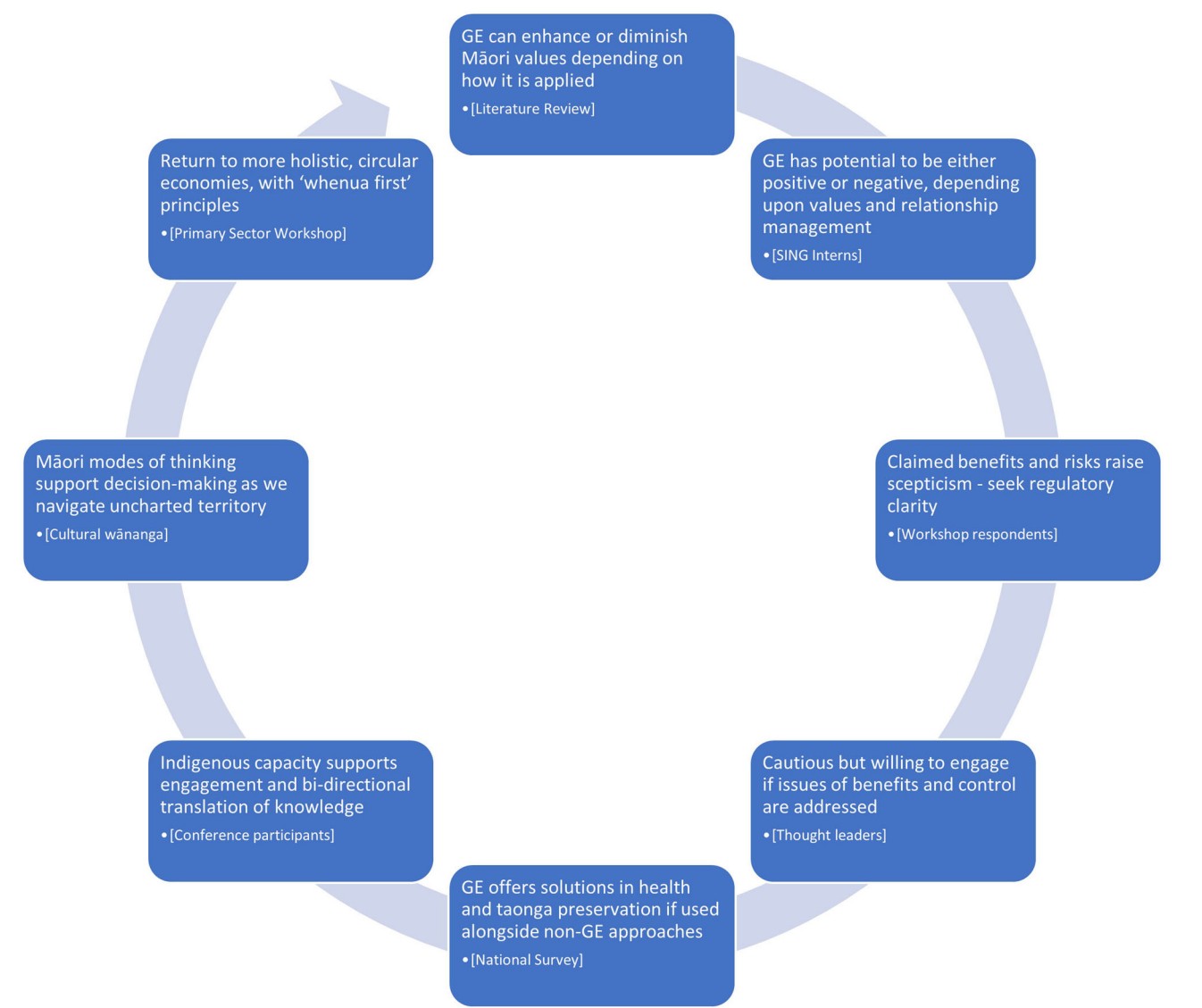

**Fig. 1 | Overview of Māori perspectives.** Method: Summary of key findings from different research activities engaging Māori in discussions about gene editing. Results: Evolving levels of comfort and willingness to engage with gene editing for specific use cases if issues of benefit and control can be addressed.

1. More nuanced definitions are required for modern gene technologies.
2. Informed public debate and clearer decision-making processes must include Māori cultural views.
3. AoNZ regulatory frameworks must align with international standards.
4. A 'risk-tiered approach where regulatory burden is commensurate with risk' would provide flexibility and technological adaptability.
5. Community engagement is critical, and
6. Capacity and capability development is necessary 'within communities, the research sector and central and local government' such that potential future opportunities can be assessed.

'One size' regulation will never fit all gene editing situations. Internationally, the EU strictly regulates gene technologies, but the UK's Genetic Technology (Precision Breeding) Bill[33], tabled in May 2022, is intended to remove barriers they deem unnecessary to research into new gene editing technology. In the US, 'DIY gene editing'[34], also known as 'bio-hacking' kits are subject only to reactionary laws (Senate Bill No.180, ch 140)[35], with some States requiring warning labels on packaging. Individual (non-State) gene-drive developers may engage in 'responsive science' with scant understanding of what that actually means[36].

The trend for domestic regulation to align with international standards represents a significant challenge for indigenous values everywhere. In an AoNZ context, monitoring processes could be created and conducted by local communities, ensuring collaborative governance over genetic research and development and validating Treaty of Waitangi principles of partnership, protection and participation.

The social complexity of technical interventions is often underestimated by scientists despite calls for greater community engagement[37]. A consistent theme emerging from the diverse elements of this research was that, for Māori values and commercial objectives to coincide, a broader,

**Table 1 | Key considerations for future regulation**

| Precautionary Component | Addressing/Incorporating: |
|---|---|
| Culture | The tikanga perspective, relative to the scientific, economic and mainstream cultures – managing cultural conflicts, identifying ambiguities, clarifying common and disparate objectives. What are the range of potential benefits and risks? |
| Context | What makes one genomic project or approach more 'valid' than another? Why is a particular project relevant/critical/etc. at a given point in time? Why (or why not) would gene technologies be considered? |
| Consequence | Identification of a continuum of reasonably anticipated outcomes (for monitoring). How to accommodate (or 'predict') unpredictable outcomes? What outcomes are/are not acceptable? |
| Certainty | From the kaitiakitanga perspective – what values are employed in determining how to quantify/qualify outcomes? What *un*certainties exist? What information is required to provide confidence in decision-making? |
| Control | Who makes what decisions, when? Across-time responsive decision-making should replace initial-stage, 'consultation'-based project sign-off. How are different values balanced/mediated? |
| Cost | What level of investment is required to integrate gene-related technologies into business operations and where to go to find this out? |
| Capacity & Capability | Community-level capability enables 'authority' in decision-making and offsets confidence issues around 'legitimacy of science'. Requirement for 'community' time and expertise to attract same funding as 'government', leading to improved capacity and consolidation of capability. |
| Compromise | Acknowledgement of the dynamic nature of decision-making and the lack of certainty about the consequences of gene editing. What non-genomic alternatives exist? |

more precautionary and context-driven approach to risk analysis is required. Table 1 identifies the key precautionary considerations necessary for robust contemporary regulation of gene editing applications in AoNZ:

Commercialisation is one of the contexts that can trigger negative responses in Māori. While not all Māori consider kaitiakitanga and commercialisation to be mutually exclusive, it is clear that commercialisation outcomes are only acceptable to Māori if Māori are actively involved in the process. This is consistent with the findings of the WAI262 Waitangi Tribunal hearing into Cultural and Intellectual Property[38]. The Primary Sector workshop indicated that commercialisation within a kaitiakitanga context was preferable, but only if both were underpinned by mātauranga and tikanga Māori. Without a focus on enhancing Māori interest or control, even robust permission and engagement are insufficient.

To achieve this, several aspects were considered important, starting with a Research, Science and Technology ("RST") ecosystem controlled by Māori, that is primarily responsive to Māori priorities, and more holistic whole-of-ecosystem analysis of the impacts of gene editing. Prioritising short-term commercial targets within existing research investment portfolios is viewed as inappropriate, the preference being for more Māori-specific pathways within technology portfolios. The commercialisation context has become more relevant with the release of the New Zealand Productivity Commission's report on innovation and frontier firms[39], which suggests reducing constraints upon innovation in the primary sector by (inter alia) reviewing the regulation of GM research.

## Conclusion

Māori perspectives on genetics have evolved since the early days of debates on genetic modification. The values articulated at that time are still relevant, but there is a more nuanced understanding which recognises that such values can be impacted positively or negatively depending upon context. This emphasis on context and how gene editing technologies were being applied was reflected in both qualitative workshops and the national survey. ABS issues pervade Māori discussions of acceptable processes and appropriate use of gene editing technology. The utility of Māori values to inform broader ethical considerations in technology-based debates was recognised and supported by both Māori and non-Māori survey respondents, providing impetus for their inclusion in future regulatory processes.

This project's outcomes both add to and echo conclusions drawn from multiple disciplines and studies. Existing regulatory models have failed to keep pace with technology and are unfit for purpose in AoNZ, as elsewhere. It is reasonable to question the ability of any conventional 'fixed' regulation drafted today to remain relevant to unknown, rapidly evolving technologies of tomorrow. Regulatory dynamism and flexibility in form, coupled with 'bottom-up' community engagement, are oft-cited aspirations for emerging policy directions. A need for international regulatory integration is obvious, but the risk is that global-focused criteria may exclude indigenous values as mere minority concerns. Diverse interests seek practical integration tools for evolving biotechnologies - this reality underpins the reasonable concerns expressed in varying ways throughout this research. A tikanga-based precautionary framework thus has potential both to safeguard specific indigenous interests and values and to more generally inform international standards in gene-tech regulation.

## Data availability
Survey data are available from the corresponding author on request.

Amanda Clark[1], Phillip Wilcox[2], Sandy Morrison[3], Debashish Munshi[4], Priya Kurian[5], Jason Mika[1,4], David Chagne [ID][6], Andrew Allan[7,8] & Maui Hudson [ID][1] [✉]

[1]Te Kotahi Research Institute, University of Waikato, Hamilton, New Zealand. [2]Department of Statistics, University of Otago, Dunedin, New Zealand. [3]Faculty of Māori and Indigenous Studies, University of Waikato, Hamilton, New Zealand. [4]Management School, University of Waikato, Hamilton, New Zealand. [5]Faculty of Arts, Law, Psychology and Social Sciences, University of Waikato, Hamilton, New Zealand. [6]Plant and Food Research, Palmerston North, New Zealand. [7]Plant and Food Research, Auckland, New Zealand. [8]Faculty of Science, University of Auckland, Auckland, New Zealand. [✉]e-mail: maui.hudson@waikato.ac.nz

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

## Acknowledgements
The research team acknowledge the funding received from the Ministry of Business, Innovation and Employment for research programme 'Turbo Breeding for New Zealand Plant Industries' (C11X1602).

## Author contributions
AC led drafting of the manuscript, PW/SM/JM contributed to data collection, reviewed and editing the manuscript; DM/PK/DC contributed to data collection, reviewed, edited, and responded to reviewer comments; AA led the overall project, reviewed manuscript, and responded to reviewer comments; MH contributed to data collection, conceptualised the paper, contributed to drafting manuscript, responded to reviewer comments.

## Competing interests
The authors declare no competing interests.
