## [Peer Review File · Communications Biology]

Reviewers' comments:

Reviewer #1 (Remarks to the Author):

I agree with the authors that "This series of described research activities represent the most comprehensive collection of 322 qualitative and quantitative data about Māori perspectives on gene editing to date". The amount of data that appears to have been compiled should be a source of much valuable insight that can still be mined. Unfortunately, this manuscript does that body of work a disservice by not being adequately used in this manuscript. The data analysis methods are not described, and are not used to support the many conclusions that are drawn. I would hope, that the authors would have used existing methods of textual analysis to derive quantitative data and then use them to substantiate their many statements. I see no quantitative analysis of any of their results, but rather the manuscript is more of an intuitive description of the results. This approach does not allow the reader to be convinced that their conclusions are in any way supported.

My other objection then is that the "findings" are not used to connect with the conclusions at the end of the paper. These seem disconnected. Therefore, the insights gained by the research don't obviously lead to the conclusions.

I have made several comments in the manuscript to provide more detailed examples of what I mean.

For the reasons I have described above, I regretfully must recommend rejection of the paper. The subject matter is highly interesting though, and I encourage the authors to write a more data-based, scientifically defensible paper.

Reviewer #2 (Remarks to the Author):

This paper attempts to answer a very specific and geographically targeted question on perceptions and viewpoints of Maori people of New Zealand toward genomics and genome editing technologies. While the output is likely of relevance to inform further communications efforts and policy approaches in very specific geographies, several points can be raised about the structure and contents of the paper.

(a) The mixing of terminology between genomics and genome editing likely creates significant confusion given genomics is much more general tools used in breeding programs while genome editing is - at the time - more limited in application and scope. It is likely adding noise to any result by conflating these two terms.

(b) The authors present several methods (events/surveys/conferences) in which they attempt to discern attitudes and perceptions of the technologies (both genomics and genome editing) but little is done to try and synthesize findings across all these events (either quantitatively or qualitatively) that can specifically inform any further actions on the political or regulatory dimensions beyond fairly high level comments to "ensure all communities viewpoints and concerns are heard" and the viewpoint that "commercializations or for profit activities" generate negative reactions.

(c) Lastly, the section delving into "developing a precautionary approach to genomics" seems to steer away from the purpose of the paper in ascertaining perspectives of certain population segment and moves into authors on views on certain policy and ideological approaches.

In summary, the conclusions and discussion need to be clearly and more tangible to inform future actions by - for example - those tasked with developing policy or community engagement schemes to help elevate the utility of this paper.

Co-Innovation Reviewers' Feedback

Reviewer 1: Remarks to Author

Reviewer remarks	Author Response
I agree with the authors that "This series of described research activities represent the most comprehensive collection of 322 qualitative and quantitative data about Māori perspectives on gene editing to date". The amount of data that appears to have been compiled should be a source of much valuable insight that can still be mined. Unfortunately, this manuscript does that body of work a disservice by not being adequately used in this manuscript. The data analysis methods are not described and are not used to support the many conclusions that are drawn. I would hope that the authors would have used existing methods of textual analysis to derive quantitative data and then use them to substantiate their many statements. I see no quantitative analysis of any of their results, but rather the manuscript is more of an intuitive description of the results. This approach does not allow the reader to be convinced that their conclusions are in any way supported. I have made several comments in the manuscript to provide more detailed examples of what I mean. For the reasons I have described above, I regretfully must recommend rejection of the paper. The subject matter is highly interesting though, and I encourage the authors to write a more data-based, scientifically defensible paper.	This manuscript describes a series of research activities that were analysed independently and brought together for the purposes of this paper. While there wasn't a formal meta-analysis across all the research activities it combines a mix of quantitative analysis (Lines 86-92; 127-138; 191; 197-212) and qualitative analysis where the n-number was so low (9, 17, etc) as to render quantitative interpretation fairly misrepresentative. This mixed methods approach combines robustness (quantitative survey) with nuance (qualitative exercises) to create the narrative for the manuscript. We have been transparent about the nature of each activity so that the reader can assess the conclusions the authors have made.
My other objection then is that the "findings" are not used to	Amendments made to the conclusion to address this comment

connect with the conclusions at the end of the paper. These seem disconnected. Therefore, the insights gained by the research don't obviously lead to the conclusions.

Reviewer 1: Specific in-text comments

Line	Critique Provided	Authors' Response
55	I'd like to see a better rationale for using these methods, especially III, V, VII and VIII, for those such as this reviewer, who is not familiar with the cultural significance of the terms used. Do they have specific cultural meanings or implications about the way these events are carried out that could be important to the reader for a more complete understanding of these activities?	III. are simply interviews. We have removed the reference to Whitimaia which was the group that conducted them. V. was a workshop targeting specific Māori with expertise and experience on the topic hence the description as thought-leaders. The use of wānanga for activity VII and hui for activity VIII reflects the need to provide a culturally appropriate frame for the discussions. VII was a workshop that explored culturally grounded sense making of the results emerging from previous activities. VIII was a meeting with key informants focused on use of gene editing in the primary sector.
57	Were there any standardized procedures for conducting the reviews? What types of literature were included? Were any criteria applied to either including or rejecting publications?	Inserted (Hudson et al, 2019) and removed the dates of the research timeframe
62	(Not directly critiqued)	Remove 'Whitimaia' from subtitle 'III' under 'Methods' and changed to 'Practitioner interviews' in Page 5 title.
85	(Not directly critiqued)	Replace 'study' with 'element'. Now reads 'Three activities were the foundation of this (previously published) research element (Hudson et al, 2019):'
87	What qualifies them as "key"?	(i) now reads 'A review of 38 peer-reviewed papers ...'; (ii) reads 'Informant interviews (n = 8); (iii) now reads 'A pilot

		survey (n = 9) with Māori stakeholders and individuals’.
88	Define please	See Line 87 above (removal of inessential word ‘key’ removes need for its definition)
95	There isn’t a quantitative backing for these conclusions. Should have used some textual analysis tools.	(Ao Māori values with bracketed definitions)
101	(Reviewer highlighted ‘genomic research and gene editing’)	Across the course of the project and the multi-level nature of the discussions, we used both terms, according to the circumstances of each study element. Given the distance between community understanding of gene editing and scientific practice we often combined without conflating the two concepts. In some meetings the genomic research was used as context for discussions on gene editing, in other cases it was the natural extension of genomics to genetic modification and gene editing.
105	‘More comfortable’ How was this defined	Cited Frontiers article - See Line 85 above, no need to repeat citation
138	How was this analysis done?	Now reads ‘...with the endorsed responses collated and thematically analysed within and across the range of scenarios.’
149-50	Is this the definition of “key”? (Highlighted description of informants)	Changed title at Line144 from ‘Key Informant Interviews’ to ‘Practitioner Interviews’; text at 145 now reads ‘Interviews with seventeen informants addressed ...’
156	Referring to ‘A strong theme emerged’ - How was this determined?	Common sense interpretation of majority response. Solution: remove ‘strong’. Now reads ‘A theme emerged around the need for trust...’
160	Quantitative support for this statement could have been provided.	Cited Paper - It is impractical to verify every reference to a given source, in successive sentences.

166	'A repeated suggestion' has been highlighted by the reviewer, but without an accompanying comment ...	
180	(Not critiqued, but an instance of 'genomics vs gene editing' issue)	Refined to read 'Their qualified willingness to engage in future genomics-based research, such as gene editing ...'
186		Now reads 'This national survey (Kathlene et al, 2021) sampled 830 respondents ...'
202	Were these yes/no questions asked during the survey? I don't understand the meaning of the phrase "79% indicated a higher level of support for pest control"	Not room to summarise paper in such detail - it has been cited should readers wish for more info. Revised to read 'a higher level of support for pest control than for other applications.'
206	What does this mean quantitatively? Did the members of these groups agree that these considerations were the three terms were the most important, based on a ranking exercise they were asked to do?	There are only two, not three terms identified and the words 'a majority of respondents ... identified' should be clear. Refined to read ' A majority of respondents ... identified taonga species as an important Māori value, followed by kaitiakitanga, consistent with findings from Research Elements I and III above. '
216	Data from the survey please	Added "The survey findings indicate that, despite continuing scepticism"
218	Is this the authors' explanation or based on data from the survey?	Opening sentence is not 'data from the survey'- it is an accepted generic understanding of the science at this stage, although subsequent sentence ('for majority of Māori surveyed') clearly refers to the survey. Reworded ' It is generally accepted that gene editing offers an increased level of control that, while not absolute, provides more defined benefits and clarity about risks. '

321-22	Indeed. The data collected must be considerable. I am disappointed that not more was done to formally analyse them, or convey to the reader what was done for analysis	Analysis, appropriate to each activity, identified themes that contribute to the narrative of the paper. Within the resources of the project, we were able to complete what has been presented here.
329		(Former Lines 329-332) Now reads: 'However, taken as a whole, the key messages reinforce a maturing and nuanced conversation that is open to exploring how gene editing might contribute towards delivering positive outcomes for Māori communities and businesses, and reiterate 'the importance of addressing the continuing influence of [historic perspectives]' (Hudson et al, 2019).
332		'Diagram 1 illustrates this breadth of views - acknowledging the potential of genetic technologies such as gene editing, providing they are employed within a framework of Māori principles and values, and a culture of bi-directional knowledge-sharing and capacity building. Recognition of potential is balanced by scepticism regarding 'control' and cost-benefit issues. Gene editing 'could go either way' from both scientific and cultural perspectives. Regulation needs to recognise that gene editing is 'uncharted territory' with potential to add to (rather than replace) existing approaches; and Māori values provide holistic modes of thinking for its navigation.'
337-38	How do the following sections tie in to the results of the research? The impact of the research results and the recommendations, for example in the Key Considerations for Future Regulation needs to be more explicitly shown.	Repositioned Commercialisation after 'Key Considerations for Future Regulation' segment and changed title to 'Balancing Commercialisation and Kaitiakitanga'. Inserted 'A consistent theme emerging from the diverse elements of this research was that a broader, more precautionary and context-driven approach to risk analysis is

		required, for Māori values and commercial objectives to coincide.'
357		'... However, taken as a whole, the key messages reinforce a maturing and nuanced conversation that is open to exploring how gene editing might contribute towards delivering positive outcomes for Māori communities and businesses, and reiterate 'the importance of addressing the continuing influence of [historic perspectives] (Hudson et al, 2019). Developing a Precautionary Approach for Genomics' Now reads 'Wider AoNZ is also undertaking the process of determining a regulatory approach for genomics. Regulation of GM technologies has long been informed by the Precautionary Principle...' Re 'adding heft' to lines 383-384, I decided (given this repositioning of sections) that all that was needed was to add 'following' to the existing text and to remove 'below', so - Changed 'The Royal Society of New Zealand conclusions below are consistent with the themes that emerged from our research activities:' to 'The following Royal Society of New Zealand conclusions are consistent with the themes that emerged from our research activities: ' ... very minor, but it works.

Reviewer 2: Remarks to Author

Reviewer Remarks	Author Response
This paper attempts to answer a very specific and geographically targeted question on perceptions and viewpoints of Maori people of New Zealand toward genomics and genome editing technologies. While the output is likely of relevance to inform further communications efforts and policy approaches in very specific geographies, several points can be raised about the structure and contents of the paper.	
(a) The mixing of terminology between genomics and genome editing likely creates significant confusion given genomics is much more general tools used in breeding programs while genome editing is - at the time - more limited in application and scope. It is likely adding noise to any result by conflating these two terms	Across the course of the project and the multi-level nature of the discussions, we used both terms, according to the circumstances of each study element. Given the distance between community understanding of gene editing and scientific practice we often combined without conflating the two concepts. In some meetings the genomic research was used as context for discussions on gene editing, in other cases it was the natural extension of genomics to genetic modification and gene editing.
(b) The authors present several methods (events/surveys/conferences) in which they attempt to discern attitudes and perceptions of the technologies (both genomics and genome editing) but little is done to try and synthesize findings across all these events (either quantitatively or qualitatively) that can specifically inform any further actions on the political or regulatory dimensions beyond fairly high level comments to “ensure all communities viewpoints and concerns are heard” and the viewpoint that “commercializations or for profit activities” generate negative reactions.	A strong ‘synthesis’ is that a precautionary approach (based on identified values) should ‘specifically inform further actions in the political or regulatory dimensions’. The paper does not claim that ‘commercialisations or for-profit activities’ generate negative reactions’ as an overall outcome. On the contrary, (per Lines 139, 176, 338-341, and 418-419), respondents explicitly acknowledged a shift from this historic stance to one which is more nuanced and contextual.
(c) Lastly, the section delving into “developing a precautionary approach to genomics” seems to steer away from the purpose of the paper in ascertaining perspectives of certain population	The paper summarises a range of discrete studies and workshops, from which aggregate we have identified a consistent theme that includes a desire for a precautionary

segment and moves into authors on views on certain policy and ideological approaches.	approach. It is important that the authors provide some thoughts on the implications of the identified perspectives on policy making. Intro now reads: 'A growing body of literature addresses the need for more agile, participatory regulation in the biotech era (Kormos et al, 2021; Hartley et al, 2022; Kjeldaas et al, 2022). There is increasing recognition of the transferability of Māori-focused approaches to other indigenous communities and society in general (Walker et al, 2019).' 2nd para of Conclusion amended to include: 'Regulatory dynamism and flexibility in form, coupled with 'bottom-up' community engagement are oft-cited aspirations for emerging policy directions. A need for international regulatory integration is obvious, but the risk is that global-focused criteria may exclude indigenous values as mere minority concerns. Diverse interests seek practical integration tools for evolving biotechnologies - this reality underpins the reasonable concerns expressed in varying ways throughout this research. A tikanga-based precautionary framework thus has potential both to safeguard specific indigenous interests and values and to more generally inform international standards in gene-tech regulation.'
--	--

Emailed overview from Editor:

Editor comments	Author response
We hope you will find the referees' comments useful as you decide how to proceed. Should further experimental data or	We thank the reviewers for their considered comments

analysis allow you to address these criticisms, we would be happy to look at a substantially revised manuscript. However, please bear in mind that we will be reluctant to approach the referees again in the absence of major revisions. In particular, please note that the following revisions would be necessary for us to contact our referees again:	
Please address all the concerns of the reviewers, including their concern that a more analytical approach should be adopted in the manuscript for the purposes of more convincingly showing that the gathered data support the stated conclusions and better synthesizing findings across the different types of methods used.	We have revised the manuscript to improve the narrative, link findings across different research activities, and more convincingly tie the data to the conclusions.
Also, from an editorial perspective, we request that you explicitly mention how this manuscript builds on the pilot study that was published previously (doi.org/10.3389/fbioe.2019.00070).	Pilot study informed the quantitative survey and provided themes that were explored in the other interviews and workshops. There is a significantly larger data set included in this manuscript.
Additionally, please note that while we feel strongly that there is a place in our journal for a revised version of the manuscript that addresses both these reviewer and editorial concerns, we are still undecided as to which content type it should be classified as. Therefore, we may end up eventually deciding that the manuscript type should be changed, for example, from a Perspective to a Comment.	Happy to amend as appropriate

Reviewers' comments:

Reviewer #3 (Remarks to the Author):

My comments relate specifically to the extent to which the authors have addressed original reviewers' remarks.

On reading the revised manuscript, I still consider there is a lack of information on the methods used to analyze the qual data. This specifically relates to data sources III, V, VI, VIII. The common phrase used "feedback included", while a potential data point, does not describe the analytical method used to draw conclusions.

I make this comment with the important caveat that there are innovative qual data analytical methods available which might not appear sufficient to quant researchers (I refer generally to reviewers in this case). Ensuring that the appropriate lens for evaluation is applied here is very important. I do not believe that traditional colonial/Northern/Western approaches to research analysis should be the measure by which this manuscript is evaluated. I believe the manuscript has a unique contribution to make to the literature and the research activities described, as a package, provides a source of valuable information for informing policy development. The one remaining weakness is an insight into how the conversations (qual research activities) were treated, how the information was used, synthesised, made sense of, etc. I would not expect to see language typically applied to quant analysis. I would however expect to see language around more qual measures of robustness, including credibility, transferability, dependability, confirmability, etc.

I anticipate the authorship team have implicit knowledge of how the data was treated, it just must be made more obvious to the reader.

I also attach some comments where I picked up a couple of typos.

Response to Reviewers' Feedback - Updated

Reviewer 1: Remarks to Author

Close quotation marks	completed
Reformat diagram to ensure symmetry	completed
Statement about policy innovation	noted
Comments from Reviewer #3 (Remarks to the Author): My comments relate specifically to the extent to which the authors have addressed original reviewers' remarks. On reading the revised manuscript, I still consider there is a lack of information on the methods used to analyze the qual data. This specifically relates to data sources III, V, VI, VIII. The common phrase used "feedback included", while a potential data point, does not describe the analytical method used to draw conclusions. I make this comment with the important caveat that there are innovative qual data analytical methods available which might not appear sufficient to quant researchers (I refer generally to reviewers in this case). Ensuring that the appropriate lens for evaluation is applied here is very important. I do not believe that traditional colonial/Northern/Western approaches to research analysis should be the measure by which this manuscript is evaluated. I believe the manuscript has a unique contribution to make to the literature and the research activities described, as a package, provides a source of valuable information for informing policy development. The one remaining weakness is an insight into how the conversations (qual research activities) were treated, how the information was used, synthesised, made sense of, etc. I would not expect to see language typically	Updated as below. The outcomes of all the research engagement activities were collated, analysed thematically, compared, and synthesised to illustrate the rich nuances of Māori perspectives on genomics and gene editing. The cultural robustness of our method was in the iterative discussion of results and collaborative sense-making across the series of activities. While this remains qualitative, many other aspects are quantitative, as replicates and numbers show. Note: The comment nature of this publication means that we are summarising as series of activities into a meta-narrative about how Māori are engaging with debates around gene editing. Two of the activities are based on published material which has been substantive descriptions of the methods but the activities identified here iii,V,VI,VIII were different types of meetings where discussions were held and notes taken. As above the material was analysed thematically (reflected in the key findings of each activity) and these were also discussed with participants are subsequent activities. As such they are culturally robust if

applied to quant analysis. I would however expect to see language around more qual measures of robustness, including credibility, transferability, dependability, confirmability, etc.

I anticipate the authorship team have implicit knowledge of how the data was treated, it just must be made more obvious to the reader.

I also attach some comments where I picked up a couple of typos.

not fully representative. When looked at alongside quantitative data you can see the consistency of the findings and transferability of the results.

REVIEWERS' COMMENTS:

Reviewer #3 (Remarks to the Author):

The authors have added a couple of additional lines to explain the method of analysis of qual data. While not detailed, I'm satisfied that this extra information does to some extent explain how the data collected was synthesized and interpreted.